# Animal-appropriate housing of ball pythons (*Python regius*)—Behavior-based evaluation of two types of housing systems

**Tina Hollandt** [1]☯*, **Markus Baur**[1‡], **Anna-Caroline Wöhr**[2‡]

**1** Auffangstation für Reptilien München e. V. (Munich Rescue Center for Reptiles), Munich, Germany,
**2** Chair of Animal Welfare, Ethology, Animal Hygiene and Animal Husbandry, Department of Veterinary Sciences, Faculty of Veterinary Medicine, LMU Munich, Munich, Germany

☯ These authors contributed equally to this work.
‡ MB and ACW also contributed equally to this work.
* frontina2001@yahoo.de

**Data Availability Statement:** All relevant data are within the paper.

**Funding:** For the research Tina Hollandt approved 3000€ from the Ingo and Waltraud Pauler Fond

## Abstract

Considering animal welfare, animals should be kept in animal-appropriate and stress-free housing conditions in all circumstances. To assure such conditions, not only basic needs must be met, but also possibilities must be provided that allow animals in captive care to express all species-typical behaviors. Rack housing systems for snakes have become increasingly popular and are widely used; however, from an animal welfare perspective, they are no alternative to furnished terrariums. In this study, we therefore evaluated two types of housing systems for ball pythons (*Python regius*) by considering the welfare aspect animal behavior. In Part 1 of the study, ball pythons ($n = 35$) were housed individually in a conventional rack system. The pythons were provided with a hiding place and a water bowl, temperature control was automatic, and the lighting in the room served as indirect illumination. In Part 2 of the study, the same ball pythons, after at least 8 weeks, were housed individually in furnished terrariums. The size of each terrarium was correlated with the body length of each python. The terrariums contained substrate, a hiding place, possibilities for climbing, a water basin for bathing, an elevated basking spot, and living plants. The temperature was controlled automatically, and illumination was provided by a fluorescent tube and a UV lamp. The shown behavior spectrum differed significantly between the two housing systems ($p < 0.05$). The four behaviors basking, climbing, burrowing, and bathing could only be expressed in the terrarium. Abnormal behaviors that could indicate stereotypies were almost exclusively seen in the rack system. The results show that the housing of ball pythons in a rack system leads to a considerable restriction in species-typical behaviors; thus, the rack system does not meet the requirements for animal-appropriate housing.

## Introduction

The ball python (*Python regius*) has been a popular terrarium-housed exotic pet for more than 30 years [1]. In Europe and North America, it is frequently bred, but also imports of wild

from the AG ARK (part of the DGHT). The money was used to buy terrariums and equipment like UV-Lamps, neon tubes and thermocontrol units https://ag-ark-1.jimdosite.com/fonds/. The funders had no role in study design, data collection and analysis, decision to publish, or preparation of the manuscript.

**Competing interests:** The authors have declared that no competing interests exist.

snakes or farmed breeds are commercially available. Due to the various breeding goals (coloration, pattern, scaleless skin), the ball python has highly variable phenotypes and thus is still one of the most frequently kept snake species. The international website "www.worldofballpythons.com/morphs/" [2] for the registration of color morphs (accessed on 18 May 2020) lists 7,221 different color shades and patterns. Although the ball python is listed in the Washington Endangered Species Act Appendix II [3] and the German directive VO EG 338/97 Appendix B [4], it is exempt from reporting requirements ("Federal Directive on Species Protection" [Bundesartenschutzverordnung] Appendix 5 regarding § 7 Section 2; [5]; thus, the number of ball pythons kept as pets in Europe and North America is speculative.

The ball python is native to West and Central Africa (Nigeria, Uganda, Liberia, Sierra Leone, Guinea, Benin, Ghana, and Togo). It mainly inhabits arid savannas with temperature extremes ranging from 16 to 43°C [6] and relative humidity ranging from 60% to 95%, with high seasonal variation due to the dry (December to March) and rainy seasons (April to November) [7, 8]. The "German Expert Report on Minimum Requirements for the Keeping of Reptiles" [9] stipulates a temperature range of 26–32°C with a nighttime reduction of 5°C. A localized heat spot (basking spot) with 38°C must be provided. During daytime, the ball python often hides in rodent burrows or abandoned termite mounds [10, 11]. These possibilities for hiding offer the snake relatively constant temperature and humidity conditions. At dusk, the ball python leaves its hiding place to forage or fulfil other needs [6, 12, 13]. Being a synanthropic species, the ball python is often found near settlements and cultivated fields, where it feeds on rodents [1]. Due to its body shape, it can be considered a ground-dwelling snake, although it can be seen at low heights on trees, sufficiently robust shrubs, or termite mounds [13]. Like almost all snakes, the ball python can swim, but its life cycle is not dependent on the presence of water bodies [12]. It uses bathing possibilities especially during the molting phase [6].

The typical housing system used for pythons is the so-called rack system. It was first designed in North America around 1992 [14, 15]. A rack system is a shelving system with individual bins arranged as drawers. In some models, the bins have individual lids, in other models, they are open on the top and close flush with the upper shelf board. All bins have ventilation holes. Rack systems usually have no lighting elements, so the ambient light provides the only illumination. Heating elements are installed per drawer level, and heating pads or heating cables are most frequently used. The heating elements should be equipped with a thermostat that prevents overheating and undercooling. Racks are available in various sizes. Most importantly, the bins should be flat. Depending on the manufacturers, the synthetic material used for the bins varies from clear acrylic glass to non-transparent plastic. Regarding the bin furnishing, several variants are available. The most used substrate is newspaper, but also rodent litter or bark mulch are used. Most variants include a hiding place and a water bowl (for drinking), in some cases arranged as a bowl with crawl space underneath. Some variants contain additional structural elements such as artificial plants, a water basin (for bathing), or tree branches. Rack housing offers the advantage of quick and complete cleaning, and little space and time are needed to accommodate and maintain many snakes. Because each animal is kept individually, precise animal monitoring is easily possible. Moreover, the sparse furnishing keeps the injury risk low. Further arguments of breeders and advocators of rack housing can be found in the relevant literature [16, 17] and include the following: the animals accept feed more readily in a rack system than in a terrarium; thus, feed refusal occurs less frequently; due to the higher feed intake, the animals grow faster, resulting in a younger breeding age; the animals reproduce more readily; accommodation in the rack system is more natural for the ball python, which in nature lives in termite mounds; the flat design of the bins is thought to cause less stress for the snake [18]; animals housed in rack systems are considerably less aggressive

[17]. Light causes stress for crepuscular and nocturnal animals—a further argument for indirect or no illumination in the rack drawers.

Arguments against rack housing are comprehensively presented in the expert report of Workgroup 8 (Pet Trade and Pet Husbandry) [19] from 19 July 2013; the workgroup comprises members of the "Veterinary Association for Animal Protection" (Tierärztliche Vereinigung für Tierschutz e. V.), the "Federal Association for Expertise on Nature, Animal, and Species Protection" (Bundesverband für fachgerechten Natur-, Tier- und Artenschutz e. V.), the "Workgroup Diseases of Amphibians and Reptiles" (Arbeitsgemeinschaft Amphibien- und Reptilienkrankheiten, a subdivision of the "German Society for Herpetology and Herpetoculture" [Deutsche Gesellschaft für Herpetologie und Terrarienkunde e. V.]), the "German Veterinarian Society" (Deutsche Veterinärmedizinische Gesellschaft e. V. [DVG]), the DVG "Study Group Zoo Animal, Wild Animal, and Exotic Animal Medicine" (DVG Fachgruppe Zootier-, Wildtier- und Exotenmedizin), the DVG "Study Group Pet Birds, Zoo Birds, Wild Birds, Reptiles, and Amphibians" (DVG Fachgruppe Zier-, Zoo- und Wildvögel, Reptilien und Amphibien), and the "Munich Rescue Center for Reptiles" (Auffangstation für Reptilien München e. V.). In the expert report, the workgroup pointed out the lacking possibility for three-dimensional locomotion due to the low height of the rack bins. Furthermore, the small space allowance leaves little room for furnishings, excluding possibilities for hiding in various places (dry, humid, elevated) and for climbing. Depending on the substrate, burrowing may also be impossible. Another concern, not directly addressed in the expert report, is illumination. At the most, rack systems allow illumination via ambient light or via an LED strip fixed to the lid of the bin. Spotlights, for example with UV light, cannot be installed.

In contrast to rack systems, terrariums have been used much longer for housing animals. In 1964, the "German Society for Herpetology and Herpetoculture" was founded in Germany [20]. A terrarium is an enclosure or a container in which various species of animals can be housed [21]. The interior climatic conditions are adjusted to the needs of the housed animal species. At least one side of the terrarium is transparent. In contrast to an aquarium, terrestrial elements and air space predominate. Due to the rapid technical developments in almost every area, today's terraristics is highly progressive. Daytime-dependent variations of temperature, lighting, and humidity can precisely be planned, simulated, and controlled. A skilled use of UV lamps, irrigation systems, and nebulizers in the terrarium allows creating a microclimate that is nearly identical to the microclimate in the natural habitat. In "good terraristics practice," the animal is provided with various elements for expressing its needs. Climbing possibilities, various hiding places, substrate for burrowing, and plants are included according to the housed animal species. Living plants not only ensure the formation of a natural microclimate but also provide structural change over time. Various types of light sources can be used for illumination. Energy efficient LED bulbs can provide basic illumination. To simulate natural sunlight for the basking spot, UV lamps of appropriate wavelengths and intensity should be selected according to the animal species. Similarly, heating elements should optimally radiate heat like the sun, i.e., from the top to the bottom.

Beyond the body of specialized literature, we found a few arguments against housing the ball python in a terrarium [16, 17], but these arguments are based on observations and have not been analyzed scientifically. According to the "German Expert Report on Minimum Requirements for the Keeping of Reptiles" of 1997, the ball python does not feel safe in a terrarium exceeding a certain height. Because this snake is a ground dweller and not a good climber, a terrarium that is too high poses the risk of the animal falling and getting injured [16]. Furthermore, due to perception of the surrounding environment (e.g., through a glass front), the snake feels threatened and often reacts very aggressively [17]. Lighting additionally stresses the ball python [17]. All these factors can lead to feed refusal, slow growth, and poor

reproduction rates in a terrarium. Moreover, growth of health hazardous bacteria and molds often occurs in a terrarium [17].

Many wild animals kept in captivity show stereotypical behaviors. A stereotypy is a repetitive, invariant behavior or movement pattern without function or goal and is often seen due to inadequate husbandry conditions [22]. Therefore, stereotypies are often considered as indicators of impaired wellbeing caused by acute or past suffering. As seen almost exclusively in circumstances of confinement [23, 24], situations can arise in which an animal is strongly motivated to show a behavior but cannot express it because the necessary circumstances are not given [25]. Endogenous and exogenous stimuli can induce a readiness to act that is displayed at varying intensity. However, a desire-consuming final action never happens [26] because the human-made environment does not allow it [27, 28]. Such a conflict situation evokes a coping strategy by which the animal seeks alternative possibilities to cope with a frustrating situation that it can neither avoid nor change. The associated action often begins with aggressive behavior, which is expressed strongly or weakly, depending on the animal species. If this behavior does not change the situation, deprivation develops. If the circumstances continue to remain unchanged, certain stimuli will lead to behavior patterns that have no function or goal. In invariant environmental conditions, these behavior patterns are shown increasingly often and manifest as a stereotypy [25]. Stereotypies can be divided into two categories. One is referred to as redirected action, whereby a behavior is directed at an inadequate object (e.g., a male tortoise may try to copulate with a shoe). The other category is the so-called vacuum activity, whereby no object is used (e.g., walk stereotypies or, as in the present study, crawl stereotypies). A walk or crawl stereotypy can be based on one of two functional areas of behavior. The behavior may represent an escape attempt or a search behavior (food, mate, other resources). An escape attempt always indicates a state of arousal along with discomfort and thus a reduced wellbeing [29].

Therefore, the housing environment should be designed in a way that always allows the animals to express their natural behavior repertoire and to cope with all arising challenges [30, 31]. Moreover, enriched housing conditions can evoke positive emotions, which cause improved wellbeing and contribute to solving behavioral problems [31]. The aim of the present study is a scientific, comparative evaluation of ball python husbandry by considering animal welfare aspects when housing these animals in a rack system or a terrarium.

## Animals, materials, and methods

The ethics committee of the veterinary department of the LMU Munich has approved the research under the number 99-20-10-2017.

### Ball python (*Python regius*)

Thirty-five ball pythons (*Python regius*) were used for this study (see Table 1). Twenty-five of them were male, nine were female, and one was juvenile of undetermined sex. Three of the pythons had been handed in by private persons, whereas the others had been confiscated from five snake keepers by authorized agencies. Thirteen of the pythons were between 3 and 18 years old. The age of the other pythons (*n* = 22) was unknown.

Following the study the snakes were kept in the Auffangstation für Reptilien e.V (rescue center for reptiles, munich) and cared for without any restrictions on their well-being. A total of 25 of these ball pythons were placed in verified private homes by 2021 (2018: 14 animals, 2019:6 animals, 2020:5 animals). The remaining 10 ball pythons will continue to be cared for at the Auffangstation für Reptilien München e.V. and are available for adoption. The new owners must fill out a questionnaire and attach pictures of the husbandry to adopt an animal

**Table 1. Characteristics of the studied ball pythons (*n* = 35).**

| Animal number | Sex | Age (years) | Length (cm) | Weight (g) | Color/pattern | Origin |
|---|---|---|---|---|---|---|
| 1 | male | unknown | 125 | 1,570 | WT | CA |
| 2 | male | unknown | 128 | 1,470 | WT | CA |
| 3 | male | unknown | 100 | 890 | M, Albino | CA |
| 4 | female | unknown | 104 | 1,400 | WT | CA |
| 5 | male | unknown | 95 | 420 | M, Albino | CA |
| 6 | male | 15 | 115 | 1,300 | WT | PP |
| 7 | male | 15 | 110 | 1,305 | WT | PP |
| 8 | male | 3 | 100 | 1,100 | M, Albino | CA |
| 9 | male | unknown | 98 | 1,000 | M, Banana Spider | CA |
| 10 | male | unknown | 100 | 1,190 | WT | CA |
| 11 | male | unknown | 110 | 970 | M, Spider | CA |
| 12 | female | unknown | 85 | 630 | WT | CA |
| 13 | female | 3 | 110 | 1,400 | M, Cinnamon | CA |
| 14 | male | unknown | 100 | 1,000 | WT | CA |
| 15 | female | 4 | 110 | 1,210 | M, Butter Spider | CA |
| 16 | male | 5 | 120 | 1,580 | M, Enchi | CA |
| 17 | female | unknown | 120 | 1,330 | M, Desert Pin | CA |
| 18 | male | 4 | 119 | 1,830 | M, Pewter Blast | CA |
| 19 | female | 4 | 120 | 1,580 | M, Pastel | CA |
| 20 | female | 7 | 115 | 1,530 | WT | CA |
| 21 | male | unknown | 120 | 1,440 | WT | CA |
| 22 | male | 4 | 125 | 1,590 | M, Phantom Bumble Bee | CA |
| 23 | male | 4 | 100 | 1,200 | M, Pewter | CA |
| 24 | male | unknown | 105 | 1,100 | M, Spider | CA |
| 25 | male | 6 | 130 | 1,690 | M, Yellow Belly | CA |
| 26 | male | unknown | 115 | 1,300 | M, Caramel | CA |
| 27 | male | unknown | 120 | 1,090 | M, Desert Ghost | CA |
| 28 | male | unknown | 148 | 2,530 | WT | CA |
| 29 | female | unknown | 135 | 2,200 | WT | CA |
| 30 | male | unknown | 110 | 1,300 | WT | CA |
| 31 | undetermined | unknown | 53 | 118 | WT | PP |
| 32 | male | unknown | 135 | 1,654 | WT | CA |
| 33 | male | 18 | 112 | 755 | WT | CA |
| 34 | male | unknown | 98 | 731 | M, Pastel | CA |
| 35 | female | unknown | 110 | 758 | WT | CA |
| **Mean ± SD** | | 7 ± 5 | 111.4 ± 16.6 | 1,233.5 ± 505.4 | | |

WT = wild type; M = morph; CA = confiscating agency; PP = private person.

from the sanctuary. If there is no or little previous knowledge, one or more personal interviews will be held at the station. The placement is exclusively in terrarium keeping with appropriate minimum dimensions and design according to their needs.

**Body weight, length, and color.** At the beginning of this study, the pythons had a body length ranging from 53 to 148 cm and a body weight ranging from 0.11 to 2.50 kg. We did not find a sex-specific length or weight distribution. Approximately half (*n* = 18) of the snakes had a color or a pattern (or both) divergent from the wild type (see Table 1).

**Feeding.** In feeding intervals of 2 weeks, the snakes were offered defrosted mice (*Mus musculus*) warmed up to body temperature. The juvenile snake (No. 31) received "hoppers"

(subadult mice), the adult snakes received adult mice, and the largest snakes received subadult rats (*Rattus rattus*). The numbers and sizes of the feeder animals were tailored to each snake based on personal experience. Seventeen pythons ate dead mice from the first feeding onward, whereas eighteen pythons refused to eat dead feeder animals despite multiple offerings during various daytimes and with simulation of prey movement. Therefore, these pythons were offered living mice from the third feeding onward, and five of them began eating. From the sixth feeding onward, living multimammate mice (*Mastomys coucha*) and living rats were offered. Seven snakes that had not accepted feed until then ate these feeder animals, but another six feed-refusing pythons did not. Because young, small guinea pigs (*Cavia porcellus*) were not available, defrosted mice were covered with pieces of guinea pig fur. With this method, all feed-refusing snakes finally ate. This specialization on only one species of feeder animal was due to the previous husbandry conditions in which the snakes were mostly fed newborn guinea pigs (source: confiscating agency).

## Housing systems

For the present study, the pythons were kept in two types of housing systems. First, they were housed in a rack system. Afterwards, they were housed in terrariums.

**Housing in the rack system.** The rack system consisted of clear acrylic polypropylene bins (70 × 40 x 16 cm LWH) with ventilation holes in the front and back sides (see Figs 1 and 2). The bins were placed precisely fitted as drawers in a shelving system consisting of a non-transparent plastic frame and boards made of oriented strand board. The back half of the bin was heated with a heating cable and pad, controlled via a thermostat (Thermo Control Pro II, Lucky Reptile). The daytime temperature from 8:00 a.m. to 8:00 p.m. was on average 28˚C (26–32˚C) at the back end measured above the heater element and on average 26˚C (27–30˚C)

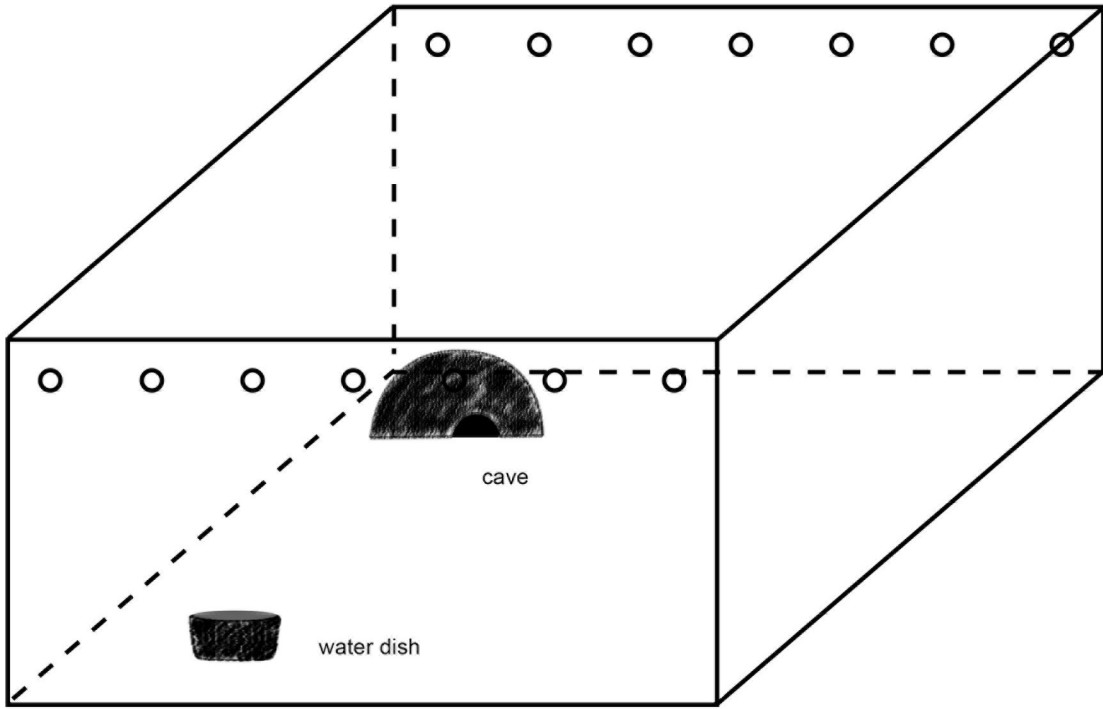

**Fig 1. Schematic view of a rack drawer.**

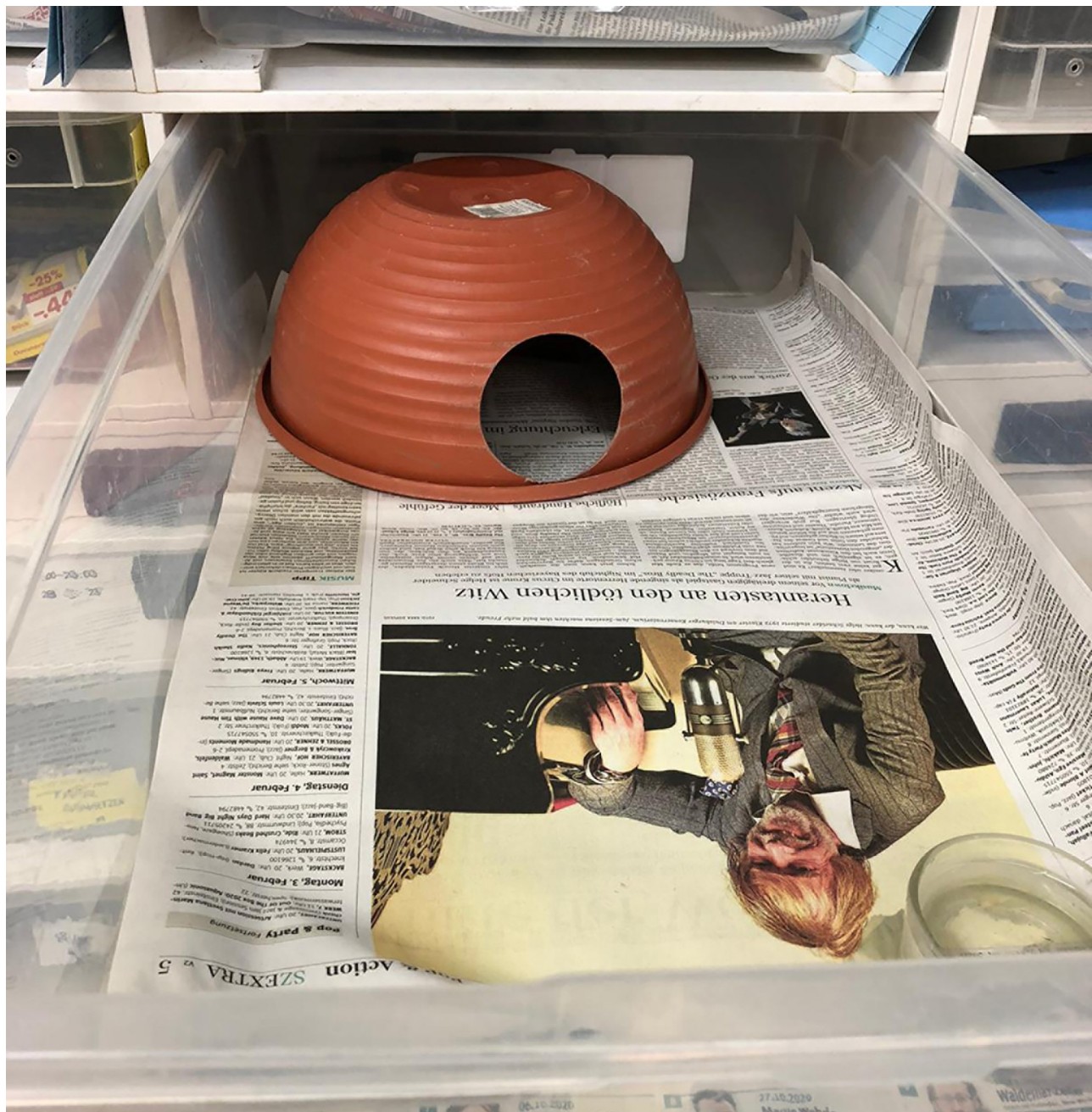

**Fig 2. Photo of a rack drawer.**

at the front end of the bin. In the time from 8:01 p.m. to 7:59 a.m., the temperature at each end was 3˚C less. The bottom was covered with newspaper. An upside-down plastic plant pot of 27 cm diameter with an entrance hole of 8 cm diameter served as hiding place. During the molting period, moist towels were put inside the hiding place. Fresh water was provided ad libitum in a bowl that was fixed to the bottom and one side of the bin with a hook-and-loop fastening strap. For the nighttime observation, one side of each drawer had a hole of 0.5 cm diameter, which allowed illuminating the drawer with red light (LED 650 nm). This wavelength lies outside the visible spectrum of the ball python [32].

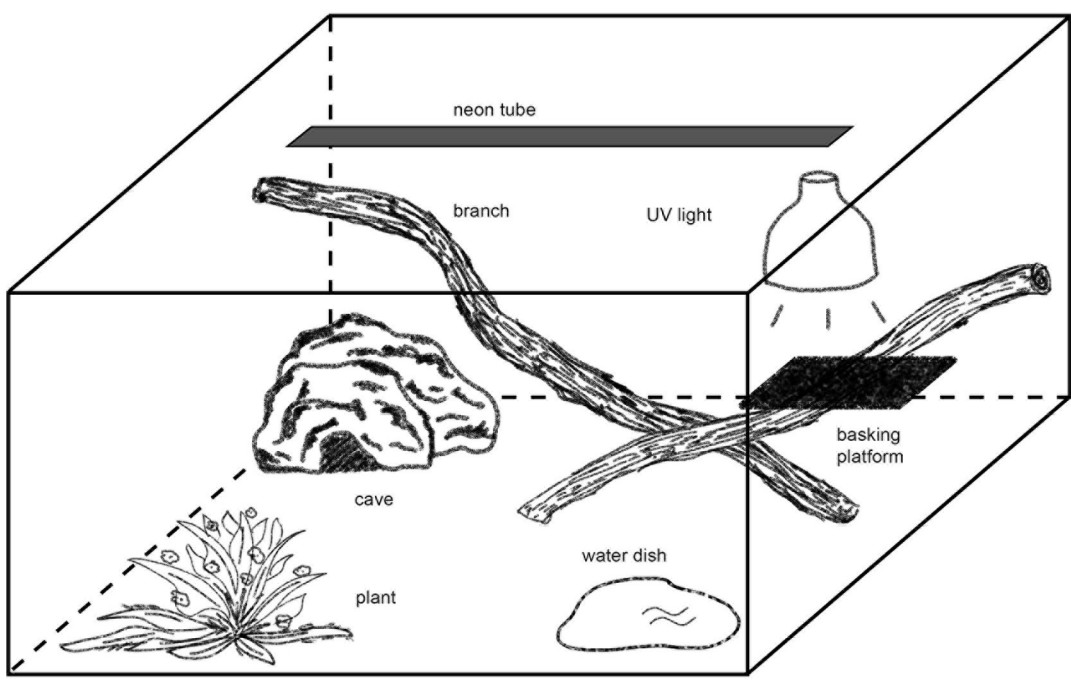

**Fig 3. Schematic view of a terrarium.**

**Housing in the terrarium.** For housing the pythons in a terrarium (see Figs 3 and 4), three sizes of terrariums were used. They met the minimum requirements for housing reptiles [9]. The smallest terrariums measured 100 × 50 × 50 cm (LWH, Size 1), the medium-sized 120 × 60 × 60 cm (Size 2), and the largest 150 × 80 × 80 cm (Size 3). Basic illumination in all terrariums was provided via a fluorescent tube (Osram 865, 6500 Kelvin; Size 1: 18 W, Size 2: 30 W, Size 3: 36 W). For protection, the tube was installed in a moisture-proof bracket. As spotlight, we used a UV lamp (Size 1: Lucky Reptile Bright Sun UV Jungle 35 W, 34 cm above the basking platform; Sizes 2 and 3: Lucky Reptile Bright Sun UV Jungle 50 W, 39 cm above the basking platform) in a protective wire case (Lucky Reptile Thermo Socket plus Reflector). The temperatures during daytime (8:00 a.m. to 8:00 p.m.) were 38°C underneath the spotlight and 25°C in the coolest area. During nighttime, the measured temperature was on average 24°C (22–26°C). The substrate was a mixture of soil (60%), sand (20%), bark mulch (15%), and loam powder (5%). In the back half, the substrate was raised to a height of 35 cm to enable the snakes to burrow. The average substrate thickness in the front half was 10 cm. Each terrarium had a hiding place like the one used in the rack system and an elevated basking platform underneath the UV lamp. Furthermore, each terrarium contained a water basin and a living plant, which was held in place by a layer of gravel. The remaining furnishings included trunks, branches, twigs, clumps of grass, roots, moss, rocks, and bark and had been collected outdoors. The arrangement of the furnishings was identical, but the use of natural materials did not allow a 100% match. For video recordings, each terrarium was illuminated with a single red LED bulb (650 nm) that was controlled with a timer.

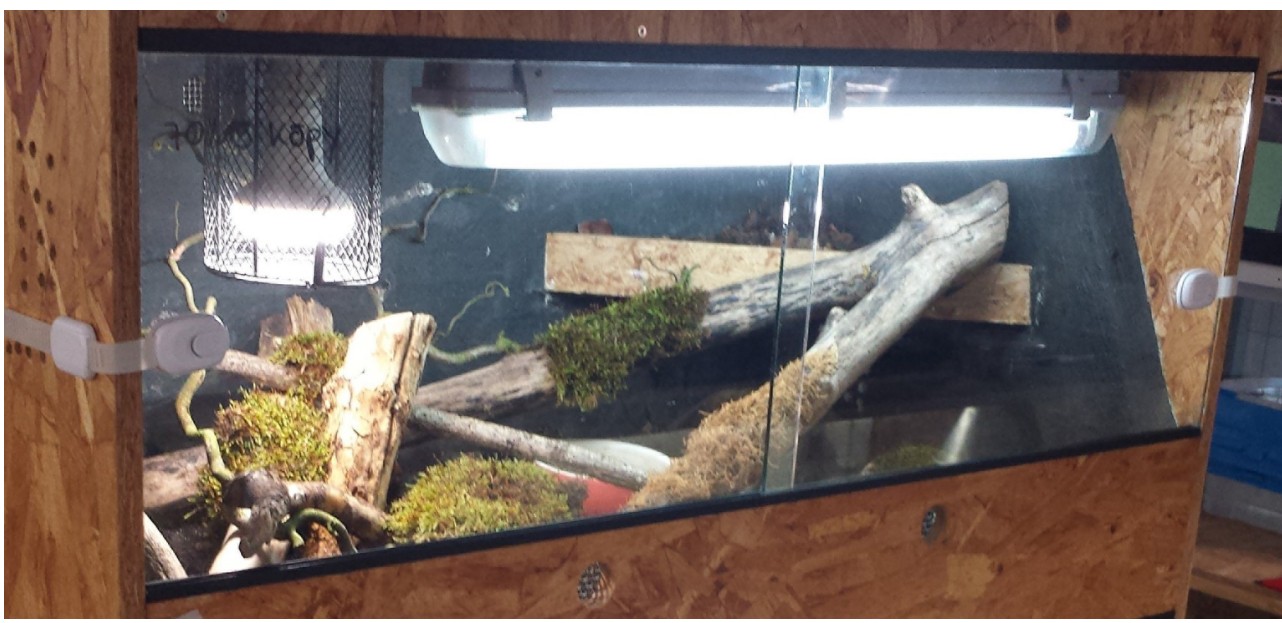

**Fig 4. Photo of a terrarium (size 1).**

## Behavior observation

All pythons were observed in the rack system and the terrarium. In the rack system, a camera (Qumox SJ 4000) was installed at the front end of the drawer and turned on for five consecutive days. All lights on the camera were covered with tape so that only the red light from the LED bulb (nighttime) and the ambient lighting in the room served as light sources. To allow an adaptation period, the camera was installed on the rack 5 days before the recording. The behavior observation began at 5:00 p.m. for 24 hours. The nighttime observation in the terrarium was also facilitated by red LED illumination. For practical reasons, the daytime observation was done without camera, although the switched-off camera remained in the terrarium. Because an ethogram for ball pythons did not exist, we created one based on the observations (see Table 2). It does not include interactions with other individuals because all pythons were single housed during the whole study. Feeding behavior is also excluded because feeding was a planned event that the individual could not control.

**Locomotion.** Behaviors were classified as locomotion when none of the other behaviors listed in Table 2 additionally occurred. "Crawling forward" includes lateral undulation, rectilinear locomotion, and a combination of both. "Moving backward" refers to movements of the whole body or of body parts. True backward crawling is not possible due to the scales, so the movement is a pushing motion facilitated by partial or complete lifting of the body. "Climbing" includes all movements during which at least half of the body does not touch the ground. "Burrowing" is an activity during which at least the head up to the eyes is burrowed in the substrate. Movements that include only the head were assessed separately.

**Exploration behavior directed at the camera.** This behavior means that the snake approaches the camera, touches it with its mouth, and probes it with its tongue.

**Comfort behavior.** Comfort behavior includes behaviors that often accompany resting behavior. "Basking" is the active visiting and staying at the basking spot, without differentiation of body positions. "Bathing" describes an active visiting of the water basin and lying in the water. Crawling through the water basin is not counted as bathing. "Resting in the hiding place with side wall contact" can be viewed as resting behavior. Lying outside of the hiding

**Table 2. Ethogram for the ball python.**

| Behavior | Abbreviation |
| --- | --- |
| Locomotion | L |
| 1. Crawling forward | L1 |
| 2. Moving backward | L2 |
| 3. Lifting the front body up | L3 |
| 4. Climbing | L4 |
| 5. Burrowing | L5 |
| 6. Moving the head | L6 |
| Exploration behavior directed at the camera | E |
| Comfort behavior | C |
| 1. Basking | C1 |
| 2. Bathing | C2 |
| 3. Resting in the hiding place with side wall contact | C3 |
| 4. Resting outside of the hiding place, not under the basking spot, coiled | C4 |
| 5. Resting outside of the hiding place, not under the basking spot, stretched out | C5 |
| Defensive behavior, aggressive behavior | A |
| Feeding behavior: drinking | F |
| Other behaviors | O |
| 1. Yawning | O1 |
| 2. Pushing the mouth against a barrier (side walls, top) | O2 |
| 3. Pathological behaviors (wobbling, stargazing) | O3 |

place, either coiled or stretched out, is a resting behavior and furthermore indicates a level of comfort in the snake because this behavior does not offer protection, in contrast to lying inside the hiding pace.

**Defensive behavior.** Defensive behavior in most cases is a sequence of behaviors. The python moves its front body into S-shaped loops and afterwards may vocalize by making a loud hissing sound. A defensive bite can occur with the mouth closed or open. All these behaviors were also recorded when they occurred individually.

**Feeding behavior.** The ethogram lists only "drinking" because the snakes could not control the timing of feeding. During drinking, the mouth (and sometimes the head up to the eyes) is submerged under the water surface in the water bowl, and water is sucked in through chewing movements.

**Other behaviors.** "Other behaviors," in contrast to the above-described ones, are not interconnected. "Yawning" is often seen after feeding but can also occur spontaneously. Another typical behavior is the crawling alongside the barriers of the enclosure whilst "pushing the mouth against side walls or the top." The pushing could be a soft touching, but it could also be strong enough to lead to temporary deformation of the mouth. "Wobbling" and "stargazing" are abnormal behaviors that mostly occur in certain color morphs (e.g., Spider) or with the onset of disease (e.g., arenavirus infection). They describe a disoriented, vibrating movement with spiraling turns or crawling on the back. These movements are often associated with a stimulus, such as the offering of feed.

## Ethics statement

Before the beginning of this study, the study design was submitted to the ethics committee of the Center for Clinical Veterinary Medicine, Faculty of Veterinary Medicine, LMU Munich, Germany. The study was approved under protocol number 99-20-10-2017.

## Behavior assessments

The behaviors were documented in 10-min intervals, resulting in a dataset of 144 behavior units per day. This assessment was done on 5 days for each housing system. For comparative data analysis, the area under the curve (AUC) was calculated. For a more precise comparison of behavior rhythms in the two housing systems, we divided the day in three periods. The presumed main activity phase (Period 1; P1) from late afternoon to early night was between 4:00 p.m. and 11:00 p.m.; it was followed by the nighttime phase (Period 2; P2) until 7:00 a.m. the next day and then the daytime phase (Period 3; P3) until dusk (3:59 p.m.).

## Statistical analysis

The collected data were first transcribed in Microsoft Excel 2007 (Microsoft Corporation, Redmond, CA, USA). For statistical analysis, we used IBM SPPS Statistics (IBM Deutschland GmbH, Ehningen, Germany) and MedCalc (MedCalc Software Ltd, Ostend, Belgium). Differences between the housing systems in the frequency of shown behaviors were determined with the t-test. Differences between the daytime periods within and between the housing systems were analyzed with the t-test and the Wilcoxon test. The level of significance was set at $p < 0.05$.

## Results

In this study, we differentiated 17 behaviors (see Table 2). Defensive or aggressive behavior (A) was never shown, nor was "moving backward" (L2). "Moving the head" (L6) was never shown as a separate movement but could be observed associated with other behavior components. Table 3 lists the relative frequency of all behaviors displayed in a 24-hour period in the rack system and the terrarium.

For eight behaviors, we found a statistically significant ($p < 0.05$) difference between the housing systems. The behavior "crawling forward" (L1) was the most frequent locomotion behavior in both housing systems. It occurred significantly ($p < 0.05$) more often in the terrarium (AUC = 21.6) than in the rack system (AUC = 9.7). "Pushing the mouth against a barrier" (O2) occurred significantly ($p < 0.05$) more often in the rack system (AUC = 15.9) than in the

**Table 3. Comparison of the relative frequency of all behaviors displayed in 24 hours in the two housing systems (rack system and terrarium).**

| Behavior | Rack system (%) | Terrarium (%) |
|---|---|---|
| Crawling forward (L1) | 7.11 ± 0.25 | 15.90 ± 0.02 |
| Lifting the front body up (L3) | 0.78 ± 0.006 | 1.15 ± 0.005 |
| Climbing (L4) | 0 | 7.00 ± 0.02 |
| Burrowing (L5) | 0 | 1.17 ± 0.01 |
| Exploration behavior directed at the camera (E) | 0.55 ± 0.005 | 0 |
| Basking (C1) | 0 | 9.90 ± 0.05 |
| Bathing (C2) | 0 | 0.90 ± 0.01 |
| Resting inside the hiding place (C3) | 53.90 ± 0.15 | 33.33 ± 0.13 |
| Resting outside of the hiding place, coiled (C4) | 11.24 ± 0.08 | 11.85 ± 0.07 |
| Resting outside of the hiding place, stretched out (C5) | 14.64 ± 0.09 | 18.64 ± 0.10 |
| Drinking (F) | 0.03 ± 0.0006 | 0.07 ± 0.0009 |
| Yawning (O1) | 0.02 ± 0.0004 | 0.02 ± 0.0004 |
| Pushing the mouth against a barrier (O2) | 11.59 ± 0.02 | 0.04 ± 0.001 |
| Pathological behaviors (O3) | 0.12 ± 0.004 | 0.03 ± 0.001 |

terrarium (AUC = 0.1). The pythons spent a large part of the day resting (C3–C5). "Resting in the hiding place" (C3) was the most frequent variant and occurred significantly ($p < 0.05$) more often in the rack system (AUC = 79.6) than in the terrarium (AUC = 50.9).

"Basking" under the UV lamp (C1), "climbing" (L4), and "bathing" (C2) occurred only in the terrarium. These behaviors could not occur in the rack system because of its structural design. "Exploration behavior directed at the camera" (E), although possible in the terrarium, was shown only in the rack system (AUC = 0.9).

We also found daytime-specific differences within and between the housing systems. In the following, P1 refers to the main activity phase from 4:00 p.m. to 11:00 p.m., P2 to the night-time phase from 11:01 p.m. to 7:00 a.m., and P3 to the early daytime phase from 7:01 a.m. to 3:59 p.m.

In the terrarium, the behavior "crawling forward" (L1; see Fig 5) was shown most frequently during P1 (AUC = 38.0) and considerably less during P2 (AUC = 5.8) and P3 (AUC = 6.9). During all periods, the values differed significantly ($p < 0.0035$) from those in the rack system (P1: AUC = 16.0; P2: AUC = 3.1; P3: AUC = 2.3). In addition, the differences between the periods were considerably smaller in the rack system than in the terrarium.

During all periods, "lifting the front body up" (L3) was observed similarly often in both housing systems. This behavior occurred most frequently during P1, both in the rack system (AUC = 0.5) and in the terrarium (AUC = 5.8). During the other two periods, it occurred less often in both housing systems (AUC = 0.4 ± 0.1).

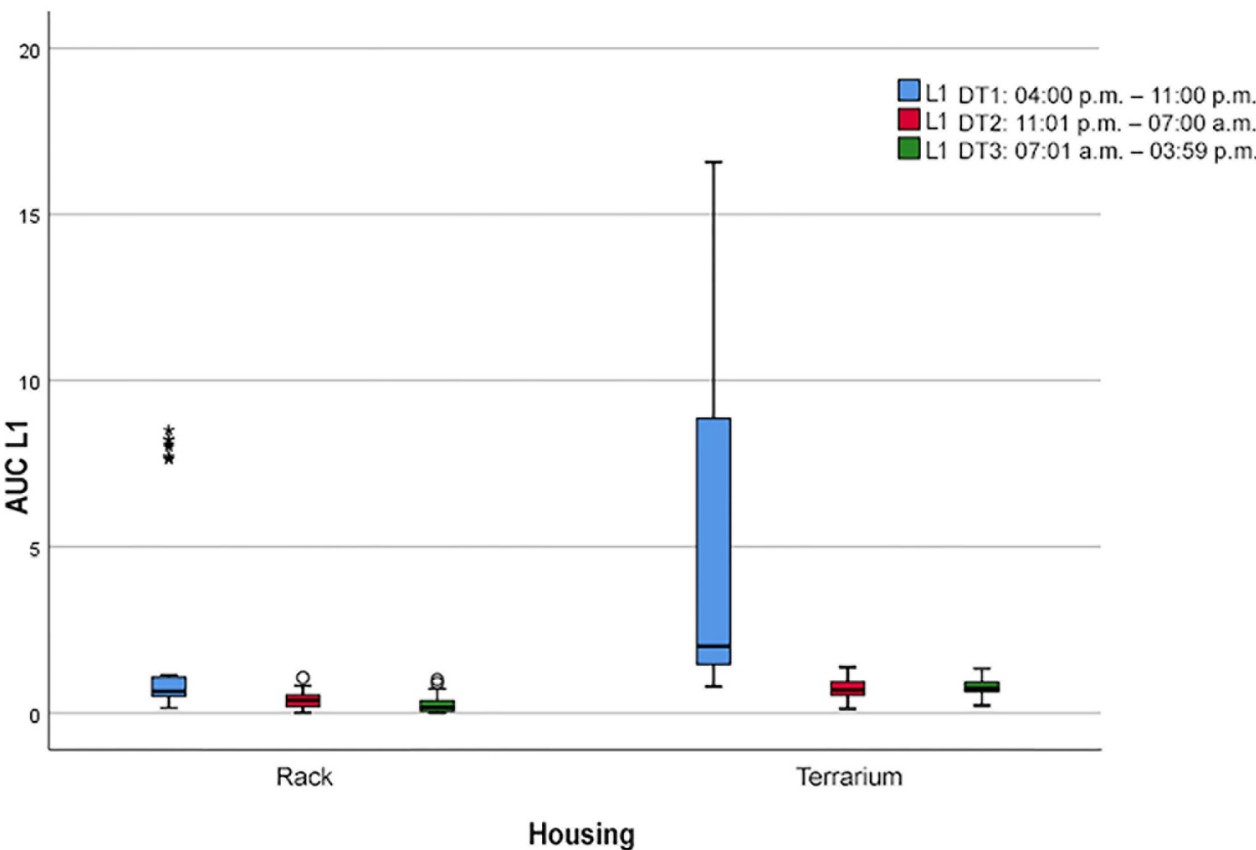

**Fig 5. Boxplot with extreme outliers** (*). Frequency of the locomotion behavior "crawling forward" (L1) during the three daytime periods (P) depending on the two housing systems ($p < 0.05$).

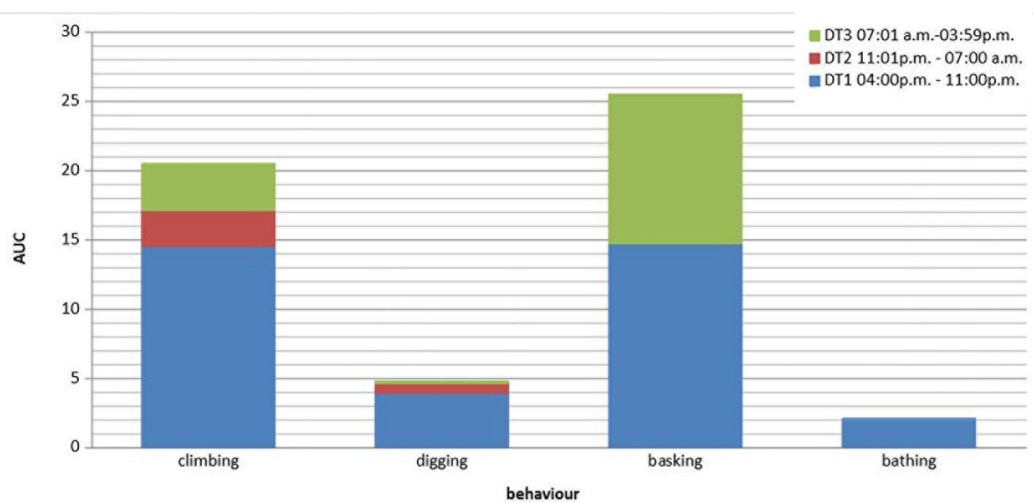

**Fig 6. Occurrence of four behaviors in the terrarium during the three daytime periods (P).**

"Climbing" (L4) behavior in the terrarium also had its activity peak during P1 (AUC = 14.5) and occurred considerably less often during the other two periods (P2: AUC = 2.6; P3: AUC = 3.4). We made similar observations (see Fig 6) for the other three behaviors that could only be shown in the terrarium. "Burrowing" (L5) occurred most frequently during P1 (AUC = 3.9), followed by P2 (AUC = 0.7) and P3 (AUC = 0.2). "Bathing" (C2) was observed most frequently during P1 (AUC = 2.1), much less during P3 (AUC = 1.0), and not at all during the nighttime period (P2). Because of the set lighting intervals, "basking" (C1) could occur only during P1 (AUC = 14.7) and P3 (AUC = 10.8). The three albinotic ball pythons were basking for on average 10 ± 2 min/day, much less than the other ball pythons, which were basking for on average 144 ± 13 min/day.

"Exploration behavior directed at the camera" (E) in the rack system occurred most frequently during P1 (AUC = 0.9) and rarely during the other two periods (P2: AUC = 0.2; P3: AUC = 0.1). It did not occur in the terrarium.

"Resting in the hiding place" (C3; see Fig 7) was most frequently observed during P1 (rack system: AUC = 63.8; terrarium: AUC = 36.4). During the other periods (P2 and P3 combined), it occurred at similar frequencies within each housing system (rack system: AUC = 29 ± 7; terrarium: AUC = 18.2 ± 1.3).

"Coiled resting outside of the hiding place" (C4; see Fig 8) was shown at similar frequencies in both housing systems during all daytime periods. We found a small behavior peak during P1 in both the rack system (AUC = 13.4) and the terrarium (AUC = 9.3). During the other two periods, this comfort behavior occurred at almost identical frequencies within each housing system (rack system: AUC = 6.2 ± 0.1; terrarium: AUC = 5.45 ± 0.45).

By contrast, "stretched-out resting outside of the hiding place" (C5; see Fig 9) in the terrarium was observed more frequently during the activity phase (P1: AUC = 18.9) and the nighttime phase (P2: AUC = 14.6) and less frequently during the early day (P3: AUC = 5.6). The frequency of this comfort behavior in the rack system during P1 and P2 (AUC = 11.6 ± 2.6) was also higher than during P3 (AUC = 5.6).

We found a considerable difference between the two housing systems for the behavior "pushing the mouth against a barrier" (O2; see Fig 10). The pythons showed this behavior significantly more often ($p < 0.05$) and almost exclusively in the rack system. In the rack system,

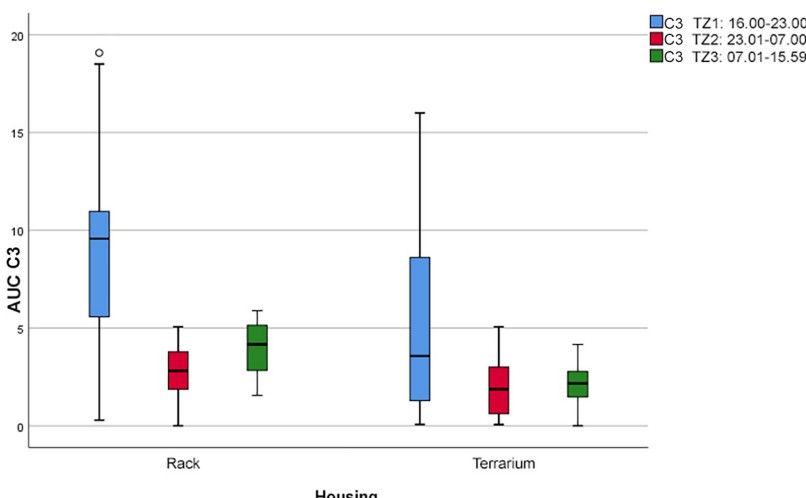

**Fig 7. Boxplot with outliers (˚).** Frequency of the comfort behavior "resting in the hiding place" (C3) during the three daytime periods (P) depending on the two housing systems ($p < 0.05$).

we furthermore observed a significant difference ($p < 0.05$) in this behavior between P1 (AUC = 33.6) and the other two periods (AUC = 4.0 ± 1.9).

A difference in "drinking" (F), "yawning" (O1), or „pathological behaviors" (O3) was not observed. The pythons showed all three behaviors sporadically during all daytime periods and in both housing systems.

## Discussion

The ball python (*Python regius*) is the most common live exported wildlife from Africa among Convention on International Trade in Endangered Species of Wild Fauna and Flora [33] listed animals. From the 3 main exporting countries, Togo, Ghana and Benin, 83,189 ball pythons were exported in 2018 according to CITES. Snakes from Togo (58,987 exported animals 2018) 93.8% came from so-called "ranching". There, pregnant females or hatchlings are collected and

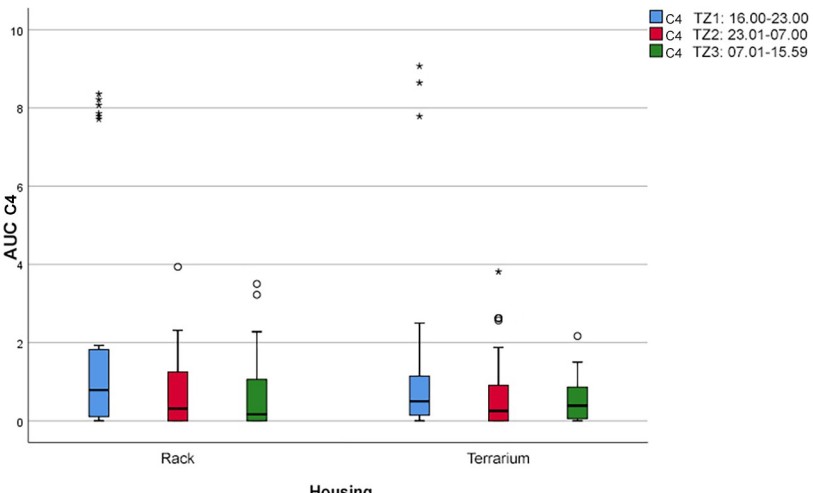

**Fig 8. Boxplot with outlier (˚) and extreme outliers (\*).** Frequency of the comfort behavior "resting outside of the hiding place, coiled" (C4) during the three daytime periods (P) depending on the two housing systems ($p > 37.92$).

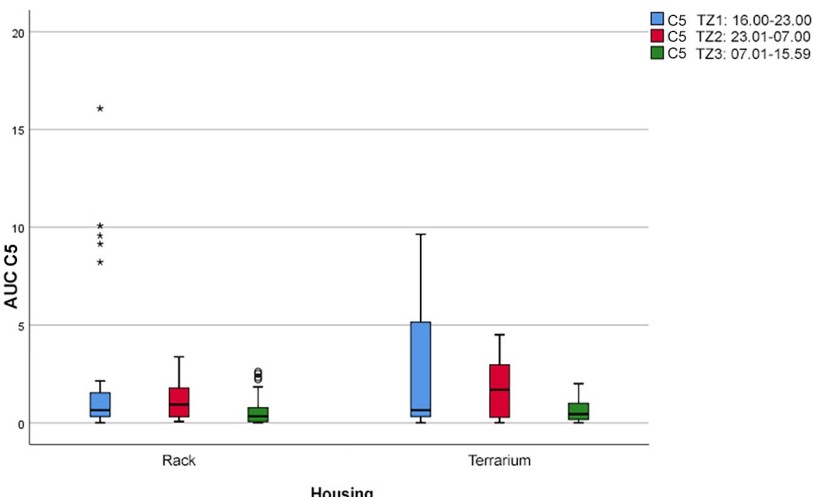

**Fig 9. Boxplot with outliers (°) and extreme outliers (∗).** Frequency of the comfort behavior "resting outside of the hiding place, stretched out" (C5) during the three daytime periods (P) depending on the two housing systems ($p > 24.38$).

brought to a farm/ranch. In theory, females are released back into the wild after laying eggs. The eggs are hatched and the hatchlings exported. In Ghana (20,952 exported animals 2018), 64.3% came from "ranching", 7.6% are declared as "captive bred animals", 28% are wild-caught. Ball pythons from Benin (3,250 exported) were 76% from "ranching"; only 2.5% of the animals came from the wild. However, there the CITES listing included 700 animals (21.5% of the export animals) that were confiscated, the origin status of the animals was not noted (sources CITES 2021 accessed on 11 of April 2021).

The main destinations of animals from the three exporting countries are the USA (83.2%) followed by Europe (8.3%) and Asia (7.8%).

According to the national pet owner survey from American Pet Products Associations 9.4 million reptiles live in American households [34]. No distinction was made among snake

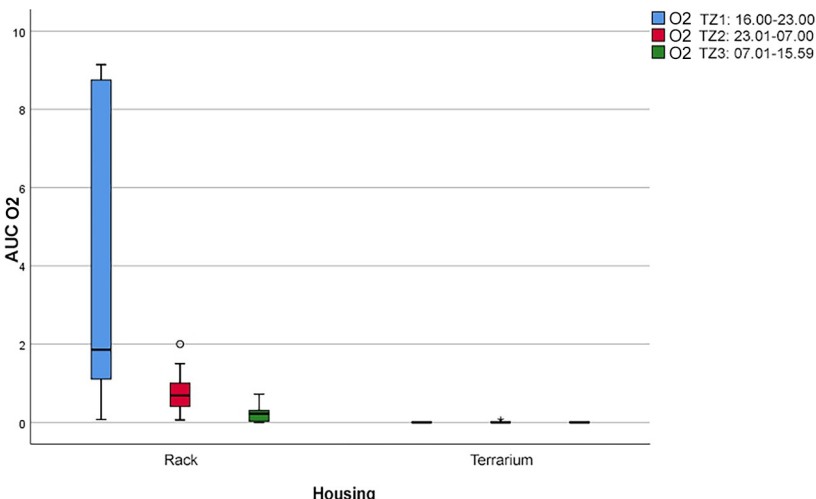

**Fig 10. Boxplot with outliers (°).** Frequency of the behavior "pushing the mouth against a barrier" (O2) during the three daytime periods (P) depending on the two housing systems ($p < 0.05$).

species. In Canada, a similar survey reported 28,000 ball pythons [35]. In the United Kingdom, an online survey by the Pet food manufacturers Association (PFMA) reported 400,000 snakes [36]. The trend from this survey made visible a doubling of snake husbandry; in the same 2019 survey, only 200,000 snakes were recorded [37]. Species were also not further identified.

In 2018, 2,498 ball pythons were imported to Germany (1,091 from Togo, 907 from the USA, 500 from Ghana) In 2019, there were 611 animals (241 from Togo, 37 from the USA). No data is available for 2020 and the current year [33].

The CITES listed ball pythons are only a small amount of kept ball pythons in every country. The ball python is easy to reproduce. Also the interest in Morphs that not found in the wild is immense. For the creation of Morphs there are a lot of wildtype/ "classic" coloured animals with morph genes but not the phenotypical expression. In Germany there is an exemption from the obligation to register the royal python. Thus, the number of snakes in private hands is not recorded ("Federal Directive on Species Protection" [Bundesartenschutzverordnung] Appendix 5 regarding § 7 Section 2; [5].

The EXOPET study (duration 2015–2018 [38], recorded 876 ball pythons kept by 292 survey participants. The husbandry facilities were racks 16.8% of the time, representing 49 keepers. Of all respondents, 288 individuals commented on the number of animals kept. Only 11 owners kept more than 13 ball pythons.

These numbers reinforce an investigation regarding housing options, namely, a rack system and a terrarium, and associated behavioral expression of the ball python. We found significant differences in the assessed behaviors depending on the housing system. The pythons in this study showed several often-underestimated behaviors (basking, climbing, bathing, burrowing), indicating the necessity for a new definition of animal-appropriate husbandry of the ball python. Although the results showed that the pythons spent most of the day resting (in the rack system: 80%, in the terrarium 64% of a 24-hour day), the way in which they rested differed between terrarium and rack system. Especially the stretched-out resting outside of the hiding place tended to occur more frequently in the terrarium. During the remaining time, the snakes also showed different frequencies in the assessed behaviors depending on the housing system.

Locomotion behaviors such as climbing and burrowing were exclusively shown in the terrarium; they could not be expressed in the rack system due to spatial and structural conditions. The ball python is considered a ground-dwelling snake [6]. However, it may occasionally crawl onto a termite mound or climb within waist-high branch wood. An animal-appropriate accommodation must therefore enable the snake to move in three-dimensional space. Burrowing and bathing were shown less often, but they are important components of the behavioral repertoire and must be facilitated for the ball python. Although bathing, a type of comfort behavior, plays only a minor role in the natural behavior of the ball python, this snake species has access to water in its natural habitat. Therefore, a large enough water basin should be provided in a housing system.

Many authors (e.g. [17]) believe that snakes do not need UV light to stay healthy. However, the behavior of the herein studied pythons clearly showed that UV light is necessary for an animal-appropriate environment that meets the needs of a ball python. The pythons actively visited the basking spot and used it daily for on average 144 min. In a preliminary study, we had found that basking spots without UV light were used significantly less than basking spots with UV light. Most ball pythons have a daily rhythm, in which they crawl to the basking spot when the light is switched on and stay there to warm up. This phase of warming up is followed by a phase of activity, which is followed by a phase of resting. Before the UV light is switched off, the snakes revisit the basking spot to warm up before dusk, when their phase of main activity begins. This natural rhythm clearly shows how the breeding of color morphs (e.g., Albino) can restrict normal behaviors. Due to their heightened light sensitivity, the albinotic pythons in our study visited the basking spot under UV light less often and for much shorter duration

(daily average: 10 min) than the pigmented pythons did. Because basking, with approx. 10% of the 12-hour light period, made up a large share in the behavior repertoire of the ball python, the question arises in how far the selective breeding of albinotic morphs represents cases of so-called torture breeding in terms of the German Animal Welfare Act [39].

The pythons showed an excessive interest in the camera only when they were housed in the rack system. This finding indicates that the ball python accepts any stimulus to express exploration behavior. Furthermore, it might explain why ball pythons easily feed and reproduce in a rack system. However, it is no evidence of animal-appropriate housing but simply indicates that the snakes use every opportunity to compensate for the lack of stimuli. In a furnished environment with many stimuli, an individual new stimulus that neither meets a basic need nor poses a clear advantage or disadvantage for the animal does not elicit interest.

In the present study, non-species-typical behavior occurred significantly more frequently ($p < 0.05$) in the rack system than in the terrarium. In rack housing, 12% of all shown behaviors were stereotypical movements, in terrarium housing, the respective frequency was less than 0.04%. The snakes crawled alongside the entire rack drawer and pushed their mouth against the sides (mostly the upper edges) and partially against the top. Several of the pythons ($n = 10$) stuck their nose through the ventilation holes and tried to widen them through burrowing movements. Because all pythons stopped showing this "mouth pushing" behavior as soon as they were transferred to a terrarium, this behavior cannot be considered a classical stereotypy, in which the behavior would be continued despite the change in circumstances [40]. However, during the rack housing period, we observed individual differences. Several pythons ($n = 9$) showed the above-described "mouth pushing" behavior on the first day of rack housing but then entered a resting state. Others ($n = 16$) initially showed a resting phase of several days, but once they started showing the "mouth pushing" behavior, they did not stop showing it for the remaining rack housing period. The remaining pythons ($n = 10$) did not show a specific pattern in the "mouth pushing" behavior. We could not find a link to any other assessed parameter. By contrast, the pathological behavior "wobbling" was not shown depending on the housing type but was exclusively shown by the color morph Spider and those resembling it ($n = 5$). Presumably, due to a deformation of the inner ear, these morphs have difficulties keeping their balance, especially in states of arousal [41].

The non-occurrence of defensive behavior in our study may be explained by the lack of a stimulus (predator, disturbance). The same applies to backward movement, which usually is observed when snakes are threatened and keep their gaze on the source of the threat while they retreat. In the present study, a threat stimulus was not given.

In summary, our study results show that based on the assessed aspects, the housing in a rack system cannot be considered an animal-appropriate accommodation for the ball python. The only animal-based advantage of rack housing is the possibility for complete and fast cleaning. This aspect can be useful for keeping sick animals or facilitating quarantine conditions. Further aspects such as the keeping of many animals in small spaces or the time-saving maintenance of these animals are in no case in the interest of the snakes. These conditions are rather reminiscent of intensive mass husbandry, in which economic aspects are considered to be of higher priority than animal welfare.

Our results do not support the argument that the ball python accepts feed more readily in a rack system than in a terrarium. With the rack system, we initially encountered difficulties in feed acceptance, but these were most likely due to the kind of offered feed. Because the snakes in both housing systems did not differ in their readiness to eat, the reason for previously reported higher growth rate in the rack system [17] is most likely a lower calory use due to reduced locomotion. Crawling forward alone made up 15% (on average) of all shown behaviors in the terrarium. In the rack system, the share of this locomotion behavior was only 7%.

Moreover, other calory-burning activities such as burrowing and climbing occurred only in the terrarium. These results suggest that the ball pythons used less energy for locomotion in the rack system and thus could invest excess calories in growth. Snakes that move little have a reduced muscle mass and tonus, as compared with snakes that can express their full behavior repertoire. Due to the reduced muscle tonus, the snakes are less able to keep their body in certain positions. A ball python that has the possibility to express all physiological movements because it lives in a furnished environment can be assumed to have stronger muscles than a ball python that lives in an unstructured and spatially restricted environment.

The statement of McCURLEY [17] that illumination is a stressor for ball pythons could be disproved in our study. If light had caused stress in the snakes, they would not have exposed themselves to it because they always had the possibility to seek shelter in a hiding place. Even the albinotic pythons, for which the duration (on average 10 min/day) of basking differed considerably from that of the pigmented pythons (on average 144 min/day), used the offered light source. For albinotic pythons, a UV lamp of low intensity should be installed. Housing with indirect illumination or in complete darkness is animal-welfare-adverse and thus not acceptable. Darkness would amplify the scarcity of stimuli in the rack system.

A terrarium must be adapted to the needs of the housed individual. For instance, the need for protection in juvenile snakes should be met with multiple hiding places and many structural elements, such as dense vegetation. The terrarium dimensions alone cannot be used to determine if a terrarium is appropriate for housing a ball python. An unstructured, large terrarium in which the animal-appropriate needs are not met is not acceptable. The terrarium should contain several hiding places, possibilities for climbing, substrate for burrowing, a large enough water basin that the snake can use for bathing, and a basking spot with UV light. The natural needs of the ball python are known and thus must be met.

## Author Contributions

**Conceptualization:** Tina Hollandt, Markus Baur.

**Data curation:** Tina Hollandt.

**Formal analysis:** Tina Hollandt.

**Funding acquisition:** Tina Hollandt, Anna-Caroline Wöhr.

**Investigation:** Tina Hollandt.

**Methodology:** Tina Hollandt.

**Project administration:** Tina Hollandt, Anna-Caroline Wöhr.

**Resources:** Tina Hollandt.

**Supervision:** Tina Hollandt, Markus Baur, Anna-Caroline Wöhr.

**Validation:** Tina Hollandt.

**Visualization:** Tina Hollandt.

**Writing – original draft:** Tina Hollandt.

**Writing – review & editing:** Tina Hollandt, Markus Baur, Anna-Caroline Wöhr.

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
