## [Decision Letter · Decision Letter 0]

24 Mar 2021

PONE-D-21-03078

Animal-appropriate housing of ball pythons (Python regius)— Behavior-based evaluation of two types of housing systems

PLOS ONE

Dear Dr. Hollandt,

Thank you for submitting your manuscript to PLOS ONE. After careful consideration, we feel that it has merit but does not fully meet PLOS ONE’s publication criteria as it currently stands. Therefore, we invite you to submit a revised version of the manuscript that addresses the points raised during the review process.

We look forward to receiving your revised manuscript.

Kind regards,

Ewa Tomaszewska, DVM Ph.D

Academic Editor

PLOS ONE

Journal Requirements:

2. In your Methods section, please include a comment about the state of the animals following this research. Were they euthanized or housed for use in further research? If any animals were sacrificed by the authors, please include the method of euthanasia and describe any efforts that were undertaken to reduce animal suffering

Additional Editor Comments (if provided):

After careful evaluation, and following reviewer assessment, I have recommended major revision.

Reviewers' comments:

Reviewer's Responses to Questions

**Comments to the Author**

1. Is the manuscript technically sound, and do the data support the conclusions?

Reviewer #1: Partly

Reviewer #2: Yes

2. Has the statistical analysis been performed appropriately and rigorously? 

Reviewer #1: No

Reviewer #2: Yes

3. Have the authors made all data underlying the findings in their manuscript fully available?

Reviewer #1: Yes

Reviewer #2: Yes

4. Is the manuscript presented in an intelligible fashion and written in standard English?

Reviewer #1: Yes

Reviewer #2: Yes

5. Review Comments to the Author

Reviewer #1: The aim of the study was to compare two housing systems of snakes: rack system and a terrarium, but in fact, these systems have a completely different use in practice. The rack system is normally propouse for large-scale snakes farming, while terrariums are mainly found in amateur breeding or in zoo exhibitions where the snakes are housed individually in optimal for species conditions, as close as possible to natural biotope. The study has no scientific significance. In fact the results were expected. It is logical and a truism that environmental conditions influence on the behawior of any organisms.

Reviewer #2: I find the work very interesting and important from the point of view of the breeder, caretakers and veterinarians. The discussed topic explains the differences in the maintenance of snakes of this species and also brings many cognitive aspects to their behavioral needs.

6. PLOS authors have the option to publish the peer review history of their article (what does this mean?). If published, this will include your full peer review and any attached files.

Reviewer #1: No

Reviewer #2: No

---

## [Author Response · Author response to Decision Letter 0]

27 Apr 2021

Manuscript: PONE-D-21-03078

Response to reviewers

Dear Reviewer 1 and 2

Thank you for reviewing the paper „Animal-appropriate housing of ball pythons (Python regius) — Behavior-based evaluation of two types of housing systems”. Thank you for your time and effort in reading and commenting on the article. Your comments and remarks have been processed accordingly and answered in the following. The page and line numbers refer to the amended manuscript. In the uploaded letters your comments are marked in italics. 

Reviewer 1 Comment to the Author

The rack system is normally propouse for large-scale snakes farming, while terrariums are mainly found in amateur breeding or in zoo exhibitions where the snakes are housed individually in optimal for species conditions, as close as possible to natural biotope.

Thank you for pointing this out. By working out numbers of animals, the distribution of the ball python in private homes could be made clearer. Unfortunately, all animals cannot be determined more precisely due to the special regulations of the countries. We assume a much higher number of unreported cases than in the presented studies; because the surveys could only be carried out with a part of the population and there is e.g. an exception of the obligation to report despite the protection status (see Germany, Page 3, 52 -58). The exhibition frequency in zoological institutions (EU currently 231, non-EU/only Eurasia 110; zoo animal list accessed on 22 of April 2021, available on: https://www.zootierliste.de/?klasse=3&ordnung=305&familie=30504&art=3030209&subhaltungen=1) also promotes the popularity of this species. Through the EXOPET study in Germany, animal numbers and husbandry types could be obtained. This showed that only 11 of the surveyed ball python keepers kept more than 13 animals. However, the information of the husbandry type showed that 49 of the 292 owners kept snakes in racks (Page 23, 495-500). Thus, it can be seen that hobby breeders are also keeping racks. As a result, it cannot be assumed that rack keeping is only practiced in large-scale farming. Through the use of social media (Facebook, Instagram), it can be concluded that rack farming is much more widespread in America than in Germany.

In fact the results were expected. It is logical and a truism that environmental conditions influence on the behawior of any organisms.

We partly agree with your comment. Environmental adaptation of the behavior of any organism is known. However, there is no scientific study of the behavior of the ball python. Thus, behavior was only revealed by non-reproducible, non-standardized observations. Through this study, the multifaceted nature of the king python's behavioral expression becomes apparent. A restriction of this behavior takes place in the investigation only in the rack. The work should serve as a scientific basis for the assessment of animal welfare. Just as cage husbandry in laying hens,crate husbandry in sows and tethering in dairy cows has become the focus of animal welfare in industrialized countries, rack husbandry of snakes must also be investigated and evaluated. Basic research is essential for this.

Reviewer 2 Comment to the Author

I find the work very interesting and important from the point of view of the breeder, caretakers and veterinarians. The discussed topic explains the differences in the maintenance of snakes of this species and also brings many cognitive aspects to their behavioral needs.

Thank you!

To support the importance of the study, we have added animal numbers (Page 22-23, 459-500). Unfortunately, a number of animals cannot be determined more precisely due to the specific regulations of the countries. We assume a much higher number of unreported cases than in the presented studies, because the surveys could only be conducted with a part of the population and there is, for example, an exception to the obligation to report despite protection status (see Germany).

---

## [Decision Letter · Decision Letter 1]

14 May 2021

Animal-appropriate housing of ball pythons (Python regius)— Behavior-based evaluation of two types of housing systems

PONE-D-21-03078R1

Dear Dr. Tina Hollandt,

We’re pleased to inform you that your manuscript has been judged scientifically suitable for publication and will be formally accepted for publication once it meets all outstanding technical requirements.

Kind regards,

Ewa Tomaszewska, DVM Ph.D

Academic Editor

PLOS ONE

Additional Editor Comments (optional):

Reviewers' comments:

Reviewer's Responses to Questions

**Comments to the Author**

1. If the authors have adequately addressed your comments raised in a previous round of review and you feel that this manuscript is now acceptable for publication, you may indicate that here to bypass the “Comments to the Author” section, enter your conflict of interest statement in the “Confidential to Editor” section, and submit your "Accept" recommendation.

Reviewer #1: All comments have been addressed

2. Is the manuscript technically sound, and do the data support the conclusions?

Reviewer #1: Partly

3. Has the statistical analysis been performed appropriately and rigorously? 

Reviewer #1: I Don't Know

4. Have the authors made all data underlying the findings in their manuscript fully available?

Reviewer #1: Yes

5. Is the manuscript presented in an intelligible fashion and written in standard English?

Reviewer #1: Yes

6. Review Comments to the Author

Reviewer #1: Comparing the husbandry of chickens, cows or other slaughter animals to the rack systems husbandry of snakes or other exotic animals is a wrong argument in my opinion. However, I agree with the argument of the need for research into large-scale exotic animal husbandry systems due to the increasing interest and demand for these animals.

It should be emphasized that the authors of the manuscript described introduction and discussion part very thoroughly and conscientiously but in material and methods - behavior observation has not been compared to any other work of this type on snakes and is rather of a contractual nature. Summing up, the research is valuable for breeders, so I would recommend it to the journal with the aims & scope that includes biology or animal husbandry.

7. PLOS authors have the option to publish the peer review history of their article (what does this mean?). If published, this will include your full peer review and any attached files.

Reviewer #1: No

---

## [Editor Report · Acceptance letter]

19 May 2021

PONE-D-21-03078R1 

Animal-appropriate housing of ball pythons (*Python regius*) — Behavior-based evaluation of two types of housing systems 

Dear Dr. Hollandt:

I'm pleased to inform you that your manuscript has been deemed suitable for publication in PLOS ONE. Congratulations! Your manuscript is now with our production department. 

Kind regards, 

on behalf of

Prof. Dr. Ewa Tomaszewska 

Academic Editor

PLOS ONE